# SARS-CoV-2 Distribution in Residential Housing Suggests Contact Deposition and Correlates with *Rothia* sp.

Victor J. Cantú,[a] Rodolfo A. Salido,[a] Shi Huang,[b] Gibraan Rahman,[b,c] Rebecca Tsai,[b] Holly Valentine,[d] Celestine G. Magallanes,[d] Stefan Aigner,[e,f,g] Nathan A. Baer,[f] Tom Barber,[f] Pedro Belda-Ferre,[b] Maryann Betty,[b,f,h] MacKenzie Bryant,[b] Martín Casas Maya,[b] Anelizze Castro-Martínez,[f] Marisol Chacón,[f] Willi Cheung,[e,f,i] Evelyn S. Crescini,[f] Peter De Hoff,[e,f,j] Emily Eisner,[f] Sawyer Farmer,[b] Abbas Hakim,[f] Laura Kohn,[k] Alma L. Lastrella,[f] Elijah S. Lawrence,[f] Sydney C. Morgan,[e] Toan T. Ngo,[f] Alhakam Nouri,[f] Ashley Plascencia,[f] Christopher A. Ruiz,[f] Shashank Sathe,[e,f,g] Phoebe Seaver,[f] Tara Shwartz,[b] Elizabeth W. Smoot,[f] R. Tyler Ostrander,[f] Thomas Valles,[b] Gene W. Yeo,[e,g] Louise C. Laurent,[f,j] Rebecca Fielding-Miller,[k] Rob Knight[a,l,m]

[a]Department of Bioengineering, University of California San Diego, La Jolla, CA, USA
[b]Department of Pediatrics, University of California San Diego, La Jolla, CA, USA
[c]Bioinformatics and Systems Biology Graduate Program, University of California San Diego, La Jolla, CA, USA
[d]Department of Obstetrics, Gynecology, and Reproductive Sciences, University of California San Diego, La Jolla, CA, USA
[e]Sanford Consortium of Regenerative Medicine, University of California San Diego, La Jolla, CA, USA
[f]Expedited COVID Identification Environment (EXCITE) Laboratory, Department of Pediatrics, University of California San Diego, La Jolla, CA, USA
[g]Department of Cellular and Molecular Medicine, University of California San Diego, La Jolla, CA, USA
[h]Rady Children's Hospital, San Diego, CA, USA
[i]San Diego State University, San Diego, CA, USA
[j]Department of Obstetrics, Gynecology, and Reproductive Sciences, University of California San Diego, La Jolla, CA, USA
[k]Herbert Wertheim School of Public Health, University of California San Diego, La Jolla, CA, USA
[l]Department of Computer Science and Engineering, University of California San Diego, La Jolla, CA, USA
[m]Center for Microbiome Innovation, University of California San Diego, La Jolla, CA, USA

Victor J. Cantú and Rodolfo A. Salido contributed equally to this work. Author order was determined alphabetically.

**ABSTRACT** Monitoring severe acute respiratory syndrome coronavirus 2 (SARS-CoV-2) on surfaces is emerging as an important tool for identifying past exposure to individuals shedding viral RNA. Our past work demonstrated that SARS-CoV-2 reverse transcription-quantitative PCR (RT-qPCR) signals from surfaces can identify when infected individuals have touched surfaces and when they have been present in hospital rooms or schools. However, the sensitivity and specificity of surface sampling as a method for detecting the presence of a SARS-CoV-2 positive individual, as well as guidance about where to sample, has not been established. To address these questions and to test whether our past observations linking SARS-CoV-2 abundance to *Rothia* sp. in hospitals also hold in a residential setting, we performed a detailed spatial sampling of three isolation housing units, assessing each sample for SARS-CoV-2 abundance by RT-qPCR, linking the results to 16S rRNA gene amplicon sequences (to assess the bacterial community at each location), and to the Cq value of the contemporaneous clinical test. Our results showed that the highest SARS-CoV-2 load in this setting is on touched surfaces, such as light switches and faucets, but a detectable signal was present in many untouched surfaces (e.g., floors) that may be more relevant in settings, such as schools where mask-wearing is enforced. As in past studies, the bacterial community predicts which samples are positive for SARS-CoV-2, with *Rothia* sp. showing a positive association.

**IMPORTANCE** Surface sampling for detecting SARS-CoV-2, the virus that causes coronavirus disease 2019 (COVID-19), is increasingly being used to locate infected individuals. We tested which indoor surfaces had high versus low viral loads by collecting 381 samples from three residential units where infected individuals resided, and interpreted the results in terms of whether SARS-CoV-2 was likely transmitted directly (e.g., touching a light switch) or indirectly (e.g., by droplets or aerosols settling). We found the highest loads where the

Address correspondence to Rebecca Fielding-Miller, rfieldingmiller@health.ucsd.edu, or Rob Knight, robknight@ucsd.edu.

The authors declare no conflict of interest.

subject touched the surface directly, although enough virus was detected on indirectly contacted surfaces to make such locations useful for sampling (e.g., in schools, where students did not touch the light switches and also wore masks such that they had no opportunity to touch their face and then the object). We also documented links between the bacteria present in a sample and the SARS-CoV-2 virus, consistent with earlier studies.

**KEYWORDS** COVID-19, RT-qPCR, Rothia, SARS-CoV-2, built-environment, environmental monitoring, isolation, quarantine, surface sampling, swab

Environmental monitoring for severe acute respiratory syndrome coronavirus 2 (SARS-CoV-2) RNA by reverse transcription-quantitative PCR (RT-qPCR) is increasingly gaining acceptance. In the Safer at School Early Alert (SASEA) (https://saseasystem.org/) project, daily surface swabbing was employed as part of an effort to detect coronavirus disease 2019 (COVID-19) cases in nine elementary schools. This study identified 89 clinically positive COVID-19 cases, with 33% of the positive cases preceded by a room-matched surface positive (1). As pandemic response measures like SASEA become more widely implemented, understanding where SARS-CoV-2 signatures will most likely be found reduces the cost and labor of surface swabbing in large facilities. Previous work focused on sampling arbitrary surfaces in isolation and congregate-care facilities, homes, and hospitals, with various detection performances obscuring which surfaces are best for monitoring COVID-19 spread (2–6). Counterintuitively, high-touch hospital surfaces expected to accumulate viral load, including door handles and patient bed rails, can yield lower SARS-CoV-2 detection rates, presumably because they are cleaned more often (7, 8).

Most microbes in the built environment come from human inhabitants (9–11). Oral, gut, and skin microbiomes of COVID-19 patients change during disease (8, 12, 13). Therefore, SARS-CoV-2 positive built environmental samples may differ in bacterial communities from SARS-CoV-2 negative samples. This has been documented in a hospital setting, with associations between SARS-CoV-2 status (detected/not detected) and both the overall microbial community and *Rothia* sp. specifically (8).

To extend these results to a residential setting and understand how SARS-CoV-2 is distributed in the living space of an infected individual, we performed environmental sampling in the apartments of three people who recently tested positive for COVID-19 (Fig. S1) while quarantined in an isolation facility. On the day of swabbing, each quarantining individual provided an anterior nares swab sample (average Cq: 29.5, 28.4, 28.6 for apartments A, B, and C, respectively). Although apartments differed in size, floor plan, and features (furniture, appliances, etc.), similar features at similar densities were swabbed across all three ($n$ = 140, 116, and 125).

Each sampled surface was swabbed twice in immediately adjacent locations: first with a swab premoistened and stored in 95% ethanol, then by a second swab premoistened and stored in a 0.5% SDS wt/vol solution (Text S1). Ethanol samples underwent 16S V4 rRNA gene amplicon (16S) sequencing, and SDS samples underwent RT-qPCR for SARS-CoV-2 detection. In the 16S sequencing, sequences were demultiplexed, quality filtered, and denoised with deblurring (14) in Qiita (15) (study ID:13957) using default parameters. The resulting feature tables were processed using QIIME2 (16).

## RESULTS

We collected 381 matched 16S and SARS-CoV-2 surface samples from the three apartments, of which 178 (47%) were positive for SARS-CoV-2 (Fig. 1) (Table S1). Apartments A and C had comparable positivity rates (53% and 61%, respectively), but apartment B was substantially lower (24%). In all three apartments, the rate of detection was highest in the bedroom (72% on average versus 47% overall). The swabbed surfaces were grouped into three categories: high-touch, low-touch, or floors. High-touch surfaces included door handles, switches, and countertops while walls, door faces, and ceiling fans were examples of low-touch surfaces. High-touch surfaces and floors had positivity rates 2 to 3 times higher than low-touch surfaces across all apartments (Table S2).

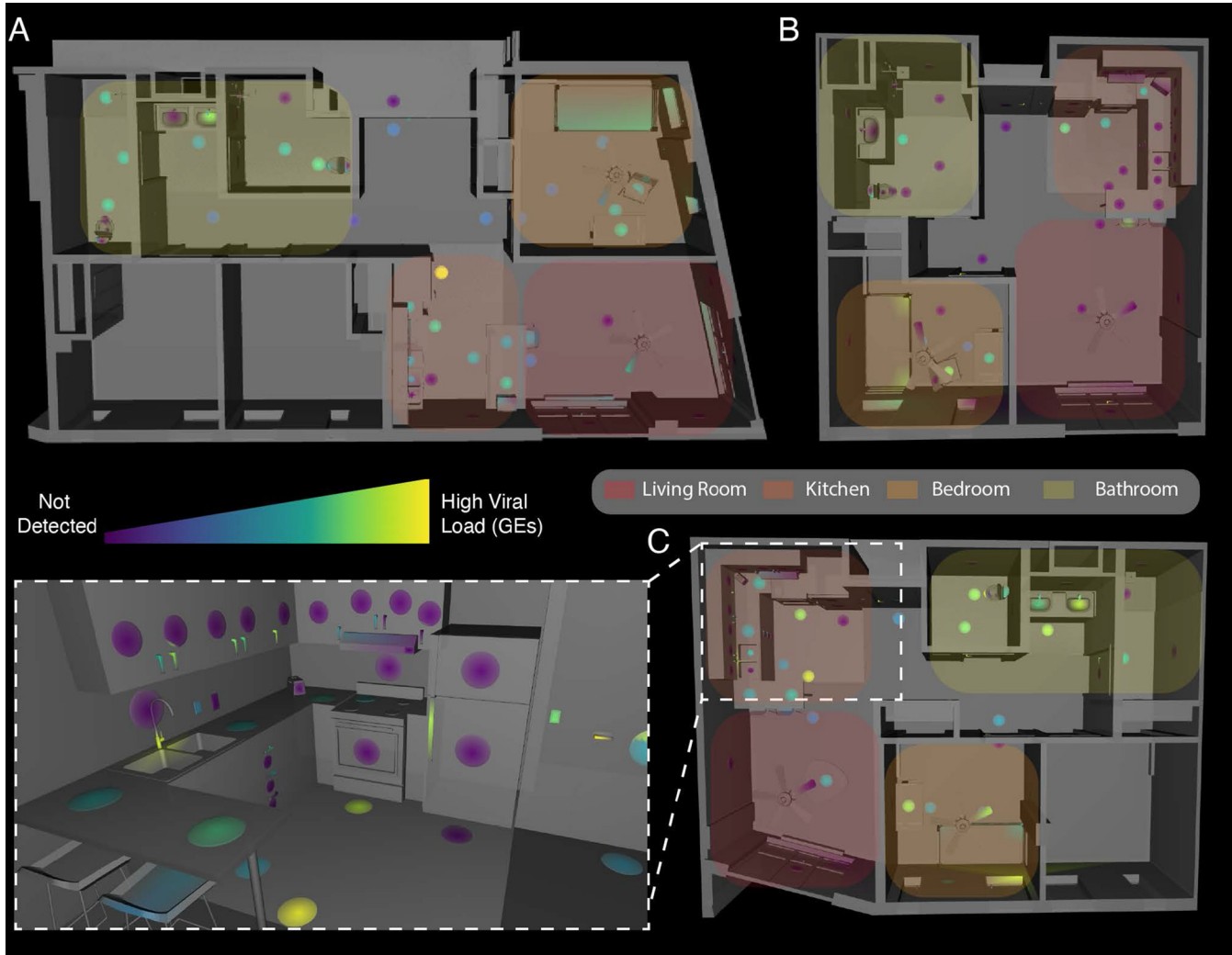

**FIG 1** Distribution of SARS-CoV-2 viral load in isolation dorm apartments. (A to C) Floor plans for each apartment highlighted where SARS-CoV-2 RNA signatures were detected. (Inset) 3D rendering of the kitchen in apartment C showing SARS-CoV-2 viral load in genomic equivalents (GEs) mapped to features in that room.

We estimated surface viral load, in viral genomic equivalents (GE's), from Cq's using published regression curves (17) and mapped resulting viral loads onto 3D renderings of each apartment. High-touch surfaces had the highest viral load across all apartments, followed by floor samples and then high-use objects (fridge, sinks, toilets, and beds) (Fig. 1). The maps for each apartment were studied to understand patterns of SARS-CoV-2 detection and deposition by room use. In the kitchens, objects with planar faces and handles, such as the refrigerator, cabinets, and drawers, revealed that only the touched handles had detectable RT-qPCR signal (Fig. 1C inset, as an example). We could not detect viral RNA on adjacent planar faces, which were presumably breathed on but not touched.

For quality control of 16S sequencing from low-biomass samples, we sequenced surface swabs from the apartments together with positive and negative controls using KatharoSeq (Text S1; Fig. S2A) (18). Of 381 samples that underwent 16S sequencing, 121 fell below the KatharoSeq threshold and were excluded (Fig. S2C). Informed by alpha rarefaction curves (Fig. S2B), the remaining samples were rarefied to 4000 features (suboperational-taxonomic-units [sOTUs] (14)), removing an additional 36 samples from the analysis. Therefore, 157 samples were excluded from downstream analyses (122 SARS-CoV-2 negative matched swabs, 35 positive) (Fig. S2C).

Bacterial alpha diversity analysis revealed a significant difference in Faith's phylogenetic diversity (Faith's PD) metric between SARS-CoV-2 detection status groupings at the whole

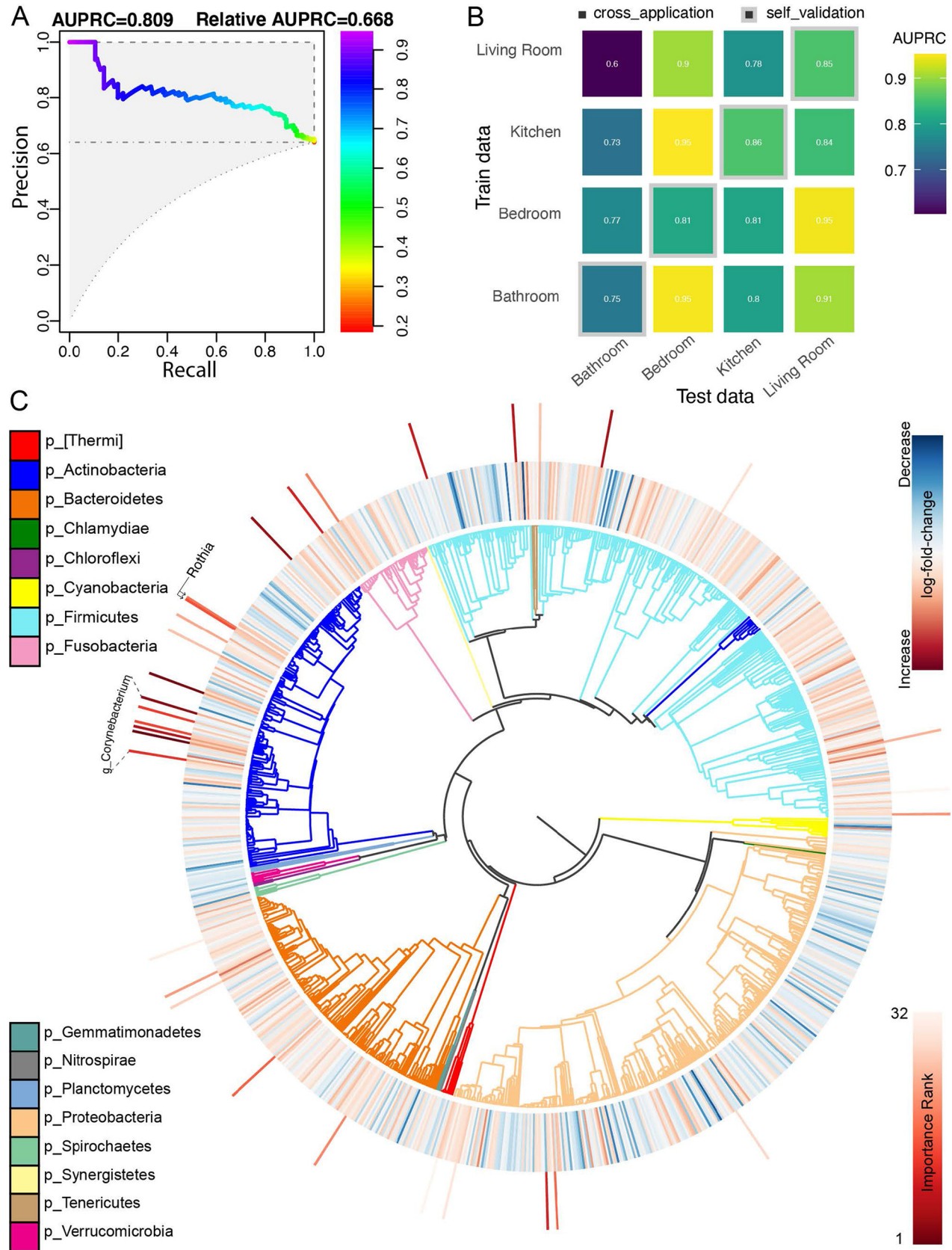

**FIG 2** (A) Area under the precision-recall curve showing the overall prediction performance of the random forest classifiers when trained on the features from two apartments and cross-validated on the remaining apartment. (B) Confusion matrix showing per-room type classifiers' performances

data set level but demonstrated limited significant differences within apartments or room types (Fig. S3). Forward stepwise redundancy analysis (RDA) using the unweighted UniFrac beta diversity metric identified four nonredundant variables of significant effect size (apartment, surface material, type of room, and SARS-CoV-2 detection status) which accounted for 45.4% of the variation in the data (Fig. S4B). Analyzed by apartment, only in apartment B did virus detection lack significant effect. When subsetting the entire data set by room type, detection status had a significant effect on variability across all rooms.

To test whether the bacterial community predicted SARS-CoV-2 status, we built a random forest classifier using rarefied sOTU data. The overall area under the precision-recall curve (AUPRC) was 0.78, suggesting a statistically significant association, but insufficiently strong to predict the SARS-CoV-2 status of a single sample from the bacterial community (Fig. 2A and B). We also applied compositionally aware, multinomial regression to our data set to identify differentially abundant microbes between SARS-CoV-2 status groups (19). Because this regression model implicitly accounts for variable sequencing depth by modeling the relative fold change of each feature in centered log-ratio (CLR) coordinates (20), we used unrarefied data as an input exclusively for this method (details in Text S1). The top 32 features identified by the random forest classifier and the ranked fold changes in feature abundance from the multinomial regression are shown in Fig. 2C. Agreeing with previously published findings, *Rothia dentocariosa* was one of the top features identified by the classifier and was relatively positively associated with SARS-CoV-2 positive samples in the regression (8, 12). Six sOTUs belonging to members of the genus *Corynebacterium* were also highly ranked as predictive for positive samples (Fig. 2C).

Our results showed that detailed spatial mapping of SARS-CoV-2 RNA abundance and associated bacterial signatures from built environment surfaces provided useful insight into potential sampling locations and associations between the viral and bacterial components of the microbiome. In the residential setting, high-touch surfaces have especially high viral loads, although confirming this with detailed spatial maps in other settings (hospitals, isolation hotels, and schools) may be useful for guiding sampling designs. However, while high-touch surfaces have higher viral loads, floors had the highest rate of positivity, effectively rendering both floors and high-touch surfaces as good candidates for detecting SARS-CoV-2 indoors. We note that the sensitivity of arbitrary single surface sampling to detect the presence of even an unmasked COVID-19 patient was low, which was evidenced in apartment B where approximately only 1 in 4 random surface samples returned a SARS-CoV-2 detection event, so multiple samples or samples from selected surfaces should be collected. Although apartment B had a considerably lower rate of positivity, trends of SARS-CoV-2 detection across indoor spaces and surface types closely mirrored those seen in the other two apartments in this study, and largely agree with other surveys of SARS-CoV-2 RNA traces in the residential setting (5, 6). These results reinforce the utility of surface monitoring as a robust, cost-effective method for locating SARS-CoV-2 signals in the environment.

Our findings also corroborated SARS-CoV-2-associated changes in the microbiome that have been previously published. *Rothia dentocariosa* has been identified across different sample types in diverse settings, although the reasons for these associations remain unclear. We also note multiple sOTUs belonging to the genus Corynebacterium predictive as of a SARS-CoV-2 detection event, in contrast to the results of another study that found Corynebacterium significantly decreased in the oral microbiome of individuals with COVID-19 (11). We hypothesize that the *Corynebacterium* signal in this study might be evidence of human skin contamination of indoor surfaces through contact (21, 22), leading to SARS-CoV-2 deposition on surfaces. It has been established that the occupants of a room contrib-

**FIG 2** Legend (Continued)
(AUPRC) when cross-applied to the remaining room types. The diagonal represents self-validation. (C) Phylogenetic tree visualization (EMPress) where the differentially abundant features between SARS-CoV-2 status groups identified by multinomial regression (Songbird) are plotted on the inner ring (red: positive fold change in SARS-CoV-2 positive group; blue: negative fold change in SARS-CoV-2 positive group) and the ranked sOTUs (top 32) identified as important by the random forest classifier are indicated on the outer ring. Leaves of the phylogenetic tree represent sOTUs relevant to the microbiome diversity and differential abundance analyses (number of sOTUs = 1047). The taxonomic classification (p_:phylum) of the sOTUs is indicated as colored branches in the phylogenetic tree.

ute to the environmental microbiota, but our findings are among the first to demonstrate that disease-associated changes in the microbiome are mirrored in the built environment.

## SUPPLEMENTAL MATERIAL

Supplemental material is available online only.

**TEXT S1**, PDF file, 0.1 MB.
**FIG S1**, JPG file, 0.3 MB.
**FIG S2**, TIF file, 0.9 MB.
**FIG S3**, TIF file, 1.5 MB.
**FIG S4**, TIF file, 0.6 MB.
**TABLE S1**, XLSX file, 0.1 MB.
**TABLE S2**, XLSX file, 0.01 MB.

## ACKNOWLEDGMENTS

This research was supported by an NIH grant (K01MH112436) to R.F.M. and the County of San Diego Health and Human Services Agency (Contract 563236).

We thank Min Yi Wu, Bing Xia, Daniel Maunder, Michal Machnicki, Bhavika K. Kapadia, and Lizbeth Franco Vargas for their support with environmental SARS-CoV-2 detection as part of the EXCITE Lab, and Gail Ackerman for her help with sequence data deposition in Qiita and EBI.

We declare no conflict of interest.

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
