## [Reviewer comments · mSystems]

SARS-CoV-2 Distribution in Residential Housing Suggests Contact Deposition and Correlates with *Rothia* sp.

Victor Cantú, Rodolfo Salido, Shi Huang, Gibraan Rahman, Rebecca Tsai, Holly Valentine, Celestine Magallanes, Stefan Aigner, Nathan Baer, Tom Barber, Pedro Belda-Ferre, Maryan Betty, MacKenzie Bryant, Martín Casas Maya, Anelizze Castro-Martínez, Marisol Chacón, Willi Cheung, Evelyn Crescini, Peter De Hoff, Emily Eisner, Sawyer Farmer, Abbas Hakim, Laura Kohn, Alma Lastrella, Elijah Lawrence, Sydney Morgan, Toan Ngo, Alhakam Nouri, R. Ostrander, Ashley Plascencia, Christopher Ruiz, Shashank Sathe, Phoebe Seaver, Tara Schwartz, Elizabeth Smoot, Thomas Valles, Gene Yeo, Louise Laurent, Rebecca Fielding-Miller, and Rob Knight

Corresponding Author(s): Rob Knight, UCSD School of Medicine

Review Timeline:

Submission Date:	November 29, 2021
Editorial Decision:	February 21, 2022
Revision Received:	March 17, 2022
Accepted:	April 20, 2022

Editor: Ileana Cristea

Reviewer(s): The reviewers have opted to remain anonymous.

Transaction Report:

DOI: <https://doi.org/10.1128/msystems.01411-21>

February 21, 2022

Prof. Rob Knight
UCSD School of Medicine
9500 Gilman Drive
MC 0602
La Jolla, CA 92093

Re: mSystems01411-21 (SARS-CoV-2 Distribution in Residential Housing Suggests Contact Deposition and Correlates with *Rothia* sp.)

Dear Rob:

Thank you for submitting your manuscript to mSystems. We have completed our review and I am pleased to inform you that, in principle, we expect to accept it for publication in mSystems. As a note, one of the reviewers who has accepted the invitation has never submitted a review. To avoid further delays, I am proceeding with the informative review that we have already received. We invite the submission of a revised manuscript that fully addresses the reviewer's comments.

Below you will find instructions from the mSystems editorial office and comments generated during the review.

Preparing Revision Guidelines

Sincerely,

Ileana Cristea

Editor, mSystems

Journals Department
Reviewer comments:

Reviewer #2 (Comments for the Author):

See attached file with review.

Summary/Overview:

This paper by Cantú et al. advances our understanding of the best surfaces for surveillance swabbing for SARS-CoV-2 in the built environment and is very timely. The data on surfaces was compelling and would improve surveillance swabbing. However, I am very skeptical of the presented results using total raw 16S read counts from each sample. Additionally, I have some concerns on the normalization methods used for the Differential Abundance and Random Forest Classification. I expanded on these points in the major revisions section. My other comments are all minor. Overall, I enjoyed reading this paper and found it very interesting.

Major Revisions:

- Line 128-129: Why were total read counts used as a proxy for biomass? Due to the compositional nature of sequencing, total 16S reads are not a good proxy for total biomass; and I highly recommend the authors either remove these results or consider replacing it (for instance, with 16S qPCR data). I recognize that adding qPCR data would add significant time to the turnaround of this manuscript; and do think removing Fig S3A and B would be a reasonable alternative.
- SI Line 443: Was any normalization on the unrarefied feature table used for the differential abundance analyses? I do not trust comparisons between samples that have not been normalized in some fashion.
- SI Line 453: What data was used to build the machine learning model? Were rarefied read counts or another normalization used? I would appreciate additional details to understand what data went into this model and to fully assess its validity.

Minor Revisions:

- Line 106: I think this should refer to “Table S1” not “Table 1”.
- Line 107-108 (and 132-134): There was a large difference in the number of positive samples in Apartments A and C compared to B. However, this wasn’t explored in the manuscript. It would strengthen the manuscript to discuss here (or later in the discussion), if there were any specific reasons that might explain this.
- Line 116: I really like the maps to visualize the sampling around each apartment. However, I think it would improve the results to add another visualization to illustrate the sentence here. Additionally, it would be useful to contextualize this with other papers on surfaces in the indoor environment in the discussion.
- Line 123-124: The use of “features” at the start of line 124 is unclear to me (suggests that it was rarefied based on taxa or something else). Should this be “rarefied to 4000 sequences” or something similar instead?
- Line 143: not sure about this method
- Line 167-169: It would be useful to cite if *Corynebacterium* is a common skin bacterium.
- Line 167: *Corynebacterium* should be italicized here.
- Figure S2: It would be more clear if in the caption the authors defined what the “+” and “-” signs stand for (I assumed that it is for SARS-CoV-2 positive or not).
- Figure S3: For clarity, I would be consistent in the use of Kruskal-Wallis and Mann-Whitney U tests (the figure caption says Mann-Whitney U, while the Alpha Diversity section of the SI uses Kruskal-Wallis).
- SI Line 471: I found the Phylogenetic Tree visualization section unclear. Does the phylogenetic tree only show the top 32 important features? My understanding from the

figure caption was that it included more than these, but just highlighted the important features in the inner and outer ring. It would be useful to expand this section to more clearly describe what was plotted and how the phylogenetic tree was created.

UNIVERSITY of CALIFORNIA, SAN DIEGO
SCHOOL OF MEDICINE

Rob Knight, Ph.D.
Professor
Department of Pediatrics
UC San Diego School of Medicine

9500 Gilman Drive MC0763
La Jolla, California 92093
Tel: (619) 543-7900
E-mail: rknight@ucsd.edu

March 16, 2022

Dear Ileana,

Thank you for overseeing the review of our manuscript. We have carefully read all of the critiques and revised our manuscript in response to their comments (responses shown in blue, manuscript changes tracked in 'Marked-Up Manuscript', and line references point to the revised manuscript). We believe the manuscript has been significantly strengthened due to these changes. Below is a point-by-point response:

Summary/Overview:

This paper by Cantú et al. advances our understanding of the best surfaces for surveillance swabbing for SARS-CoV-2 in the built environment and is very timely. The data on surfaces was compelling and would improve surveillance swabbing. However, I am very skeptical of the presented results using total raw 16S read counts from each sample. Additionally, I have some concerns on the normalization methods used for the Differential Abundance and Random Forest Classification. I expanded on these points in the major revisions section. My other comments are all minor. Overall, I enjoyed reading this paper and found it very interesting.

We thank the reviewer for their time and for carefully reviewing our manuscript. We have taken their constructive comments and edited our manuscript appropriately.

Major Revisions:

- Line 128-129: Why were total read counts used as a proxy for biomass? Due to the compositional nature of sequencing, total 16S reads are not a good proxy for total biomass; and I highly recommend the authors either remove these results or consider replacing it (for instance, with 16S qPCR data). I recognize that adding qPCR data would add significant time to the turnaround of this manuscript; and do think removing Fig S3A and B would be a reasonable alternative.

We decided to remove the total read count observations (Sup. Fig. S3A-B) as recommended, and instead focused our alpha diversity analysis on differences in Faith's phylogenetic diversity between different sample groupings. (Lines 136 - 138) (revised Sup. Fig. S3).

- SI Line 443: Was any normalization on the unrarefied feature table used for the differential abundance analyses? I do not trust comparisons between samples that have not been normalized in some fashion.

The multinomial regression method applied is appropriate for unrarefied compositional data; it employs a centered log-ratio transformation of the feature space. We included explicit discussion of the centered log-ratio transformation, and relevant references both in the main text (Lines 149-154) and the supplementary information (SI Lines 482-487).

- SI Line 453: What data was used to build the machine learning model? Were rarefied read counts or another normalization used? I would appreciate additional details to understand what data went into this model and to fully assess its validity.

Random Forest machine learning models were trained on rarefied feature tables (same feature tables described in lines 127 - 134 and used for the microbiome diversity analyses). We have made this explicit in the main text (Line 147) and supplementary information (SI Lines 495-502).

Minor Revisions:

- Line 106: I think this should refer to “Table S1” not “Table 1”.

We have changed this, and appreciate the correction.

- Line 107-108 (and 132-134): There was a large difference in the number of positive samples in Apartments A and C compared to B. However, this wasn't explored in the manuscript. It would strengthen the manuscript to discuss here (or later in the discussion), if there were any specific reasons that might explain this.

Unfortunately, we were not able to identify a verifiable explanation for the lower rate of detection for Apartment B, as this was outside of the scope of our experimental design. However, we expanded the discussion concerning the results of Apartment B, highlighting that the detection events in this apartment closely mirrored those seen in the other 2 apartments (Apartments A & C), and in the literature (Lines 173-176).

- Line 116: I really like the maps to visualize the sampling around each apartment. However, I think it would improve the results to add another visualization to illustrate the sentence here. Additionally, it would be useful to contextualize this with other papers on surfaces in the indoor environment in the discussion.

We included an additional supplementary table (Sup. Table S2) to summarize the observations drawn from the 3D maps related to high-touch vs low-touch surfaces. We also described trends surrounding rates of positivity across these different types of surfaces (high-touch, low-touch) and floors in the main text (Lines 111-115), and contextualized this with references to similar results in the literature (Lines 173-176). We thank the reviewer for the suggestion.

- Line 123-124: The use of “features” at the start of line 124 is unclear to me (suggests that it was rarefied based on taxa or something else). Should this be “rarefied to 4000 sequences” or something similar instead?

We have included a parenthetical description of the “features”, with relevant citation (Line 132).

- Line 143: not sure about this method

We have expanded on the description of this method, which is appropriate for compositional data and has proven to outcompete other popular differential abundance methods in microbiome analyses, in the main text (Lines 149 - 154) and supplementary information (SI Lines 483-484).

- Line 167-169: It would be useful to cite if *Corynebacterium* is a common skin bacterium. We have included relevant references that list *Corynebacterium* as a common human skin microbe. (Lines 185-187).

- Line 167: *Corynebacterium* should be italicized here.
We have corrected this.

- Figure S2: It would be more clear if in the caption the authors defined what the “+” and “-” signs stand for (I assumed that it is for SARS-CoV-2 positive or not).
This is a great suggestion, and we have clarified that “+” = SARS-CoV-2 positive, “-” = SARS-CoV-2 negative.

- Figure S3: For clarity, I would be consistent in the use of Kruskal-Wallis and Mann-Whitney U tests (the figure caption says Mann-Whitney U, while the Alpha Diversity section of the SI uses Kruskal-Wallis).
We appreciate the correction, and have clarified in the Supplementary Information (SI Lines 467-468). Faith’s phylogenetic diversity comparisons across different sample groupings were done with Mann-Whitney U tests.

- SI Line 471: I found the Phylogenetic Tree visualization section unclear. Does the phylogenetic tree only show the top 32 important features? My understanding from the figure caption was that it included more than these, but just highlighted the important features in the inner and outer ring. It would be useful to expand this section to more clearly describe what was plotted and how the phylogenetic tree was created.

We appreciate this critique. We have clarified the plotted elements in the Figure 2 legend, and expanded the description of the generation of the phylogenetic tree visualization in the Supplementary Information (SI Lines 514-521).

Sincerely,

Rob Knight (on behalf of all authors)

Rob Knight, Ph.D.

Director, Center for Microbiome Innovation

Professor, Department of Pediatrics, Bioengineering, and Computer Science and Engineering, UC San Diego

April 20, 2022

Prof. Rob Knight
UCSD School of Medicine
9500 Gilman Drive
MC 0602
La Jolla, CA 92093

Re: mSystems01411-21R1 (SARS-CoV-2 Distribution in Residential Housing Suggests Contact Deposition and Correlates with *Rothia* sp.)

Dear Prof. Rob Knight:

Thank you for submitting your revised manuscript and for addressing the reviewer's concerns.

Your manuscript has been accepted, and I am forwarding it to the ASM Journals Department for publication. For your reference, ASM Journals' address is given below. Before it can be scheduled for publication, your manuscript will be checked by the mSystems production staff to make sure that all elements meet the technical requirements for publication. They will contact you if anything needs to be revised before copyediting and production can begin. Otherwise, you will be notified when your proofs are ready to be viewed.

Publication Fees:

We recognize that the video files can become quite large, and so to avoid quality loss ASM suggests sending the video file via <https://www.wetransfer.com/>. When you have a final version of the video and the still ready to share, please send it to mSystems staff at mSystems@asmusa.org.

For mSystems research articles, if you would like to submit an image for consideration as the Featured Image for an issue, please contact mSystems staff at mSystems@asmusa.org.

Sincerely,

Ileana Cristea
Editor, mSystems

Journals Department
Sup. Fig. S2: Accept
Sup. Fig. S4: Accept
Supplemental Materials and Methods: Accept
Sup. Fig. S3: Accept
Supplemental Table S2: Accept
Supplemental Table S1: Accept
Sup. Fig. S1: Accept